# Physiological growth of ocular axial length among Chinese children and teenagers: A 6-year cohort study

Yanxian Chen [1,2], Xiaohu Ding[3], Ruilin Xiong[3], Jian Zhang[3], Fan Song[1], Ziwei Zhao[1], Mengying Lai[1], Yangfa Zeng[3]*, Mingguang He[1,2,3,4]*

1 School of Optometry, The Hong Kong Polytechnic University, Kowloon, Hong Kong SAR, 2 Research Centre for SHARP Vision (RCSV), The Hong Kong Polytechnic University, Kowloon, Hong Kong SAR, 3 State Key Laboratory of Ophthalmology, Zhongshan Ophthalmic Center, Sun Yat-sen University, Guangzhou, People's Republic of China, 4 Centre for Eye and Vision Research (CEVR), Shatin, Hong Kong SAR

* zengyangfa@gzzoc.com (YZ); mingguang.he@polyu.edu.hk (MH)

## Abstract

To investigate the pattern and threshold of physiological growth, defining as axial length (AL) elongation that results in little refraction progression, among Chinese children and teenagers, a total of 916 children aged between 7 and 18 years from a 6-year longitudinal cohort study were included for analysis. Ocular biometry, cycloplegic refraction and demographic data were obtained annually. Physiological growth was calculated based on myopic progression and Gullstrand eye model, respectively. The annual change in AL was found to be significantly smaller in the persistent emmetropia (PE) group compared to the incident myopia (IM) and persistent myopia (PM) group at all ages (all P < 0.05). In children with non-progressive myopia, there was observed axial elongation ranging from 0.17 to 0.23 mm/year between the ages of 9 and 12. This growth rate persisted at approximately 0.10 mm/year beyond the age of 12. While the compensated AL growth calculated using Gullstrand model was only 0.02 to 0.15 mm/year at age of 9–12, and decreased to around 0 mm/year after age of 12. For children aged 7–9 years, the cutoff point for AL growth to distinguish between progressive myopia and non-progressive myopia was 0.19 mm/year. These findings indicate a notable disparity between the thresholds of physiological growth calculated using myopic progression and Gullstrand eye model. This observation suggests that when formulating effective myopia control strategies, consideration should be given to different calculation methods when applying physiological AL growth as a starting point or target.

## Introduction

With the rapid increase in myopia prevalence, myopia control poses a significant challenge to the public health care system in the coming decades [1,2]. Several interventions, such as low dose atropine eye drops, orthokeratology and defocus spectacles, have been developed to control myopia progression, showing a reduction of axial elongation ranging from 0.2 to

**Data availability statement:** All relevant data are within the manuscript and its Supporting information files.

**Funding:** The studies were supported by PolyU - Rohto Centre of Research Excellence for Eye Care (Collaborative) (P0046333), and Start-up Fund for RAPs under the Strategic Hiring Scheme (P0048638). We thank the InnoHK HKSAR Government for providing valuable supports. The funders had no role in study design, data collection and analysis, decision to publish, or preparation of the manuscript.

**Competing interests:** The authors have declared that no competing interests exist.

0.36 mm per year compared to untreated myopic eyes, particularly in the initial years of treatment [3–5]. However, the threshold of effectiveness is often challenging to estimate, as current approaches rely on data from control groups. Therefore, assessing treatment effectiveness may benefit from considering physiological growth in myopic eyes, which is the axial elongation resulting in minimal change in refraction.

Axial length (AL) growth plays a crucial role in development and maintenance of optimal vision. Early studies have reported that axial length typically undergoes rapid growth during the first two years of life, contributing to visual maturation and emmetropization, which allows achieving clear vision on the retina without the need for corrective lenses [6–8]. Subsequently, the growth rate of AL decelerates with age, signifying a shift toward maintaining a stable refractive state. Physiological growth of AL is not limited to childhood, as it has been observed to continue even after reaching adulthood. Several investigations have reported a gradual elongation of the eye at an approximate rate of 0.06 mm/year throughout adulthood [9,10]. This ongoing growth may be compensated by changes in other ocular components resulting in no change in refractive error [11].

However, myopia involves elongated AL due to a combination of excessive and physiological growth, making it challenging to accurately distinguish between these components. Several studies tried to dissect the physiological growth of AL by using AL growth in emmetropic eyes to be a treatment target for myopic eyes [12,13]. However, there is considerable variation in the reported AL growth values for emmetropes in different studies. For example, research in Chinese children reported an annual growth of 0.33 mm/year at the age of 7 and 0.15 mm/year at the age of 12 [14], while a meta-analysis suggested values of 0.18 mm/year at the age of 7 and 0.03 mm/year at the age of 12 for East Asian children [15]. Defining physiological growth based on myopia progression, a study conducted in Shanghai reported a growth rate of 0.25 mm/year at the age of 7 in non-progressive myopes and proposed a cutoff of 0.20 mm/year, regardless of age [16]. Another study by Mutti et al. proposed a calculation method that could estimate the compensatory growth, which was significantly lower than that observed in non-progressive myopes [17]. Tang et al. developed a machine learning model to analyze physiological AL elongation. Their findings demonstrated that at the age of 6, the axial elongation ranged from 0.060 to 0.089 mm/year, considering various cornea power values [18]. However, there is currently no research that directly compares these different methods for evaluating physiological growth.

Therefore, this study seeks to bridge this research gap by investigating the physiological growth of axial length in a population of individuals with varying refractive states. By using different methods to determine physiological growth, we aim to establish a clearer distinction between physiological and excessive axial elongation, contributing to a better understanding of ocular development and refractive errors.

## Methods

### Study populations

The cohort from the Guangzhou Twins Eye Study (GTES) with 6-year follow-up was included in this study. The methods of data collection has been reported elsewhere [19,20]. In brief, the GTES cohort including over 1200 pairs of twins and their parents or siblings living in Guangzhou, China since 2006. The subjects involved in this study were recruited from 11 July 2006 to 29 August 2010. The baseline age of participants ranged from 7–15 years, and annual follow-up visits were conducted. Detailed follow-up information was illustrated in S1 Fig. Data of the first-born twin was analyzed in the current study.

Ethical approval was obtained from the ethics committees at the Zhongshan University Ethical Review Board, Zhongshan Ophthalmic Center. All examinations were conducted in

accordance with the Tenets of the World Medical Association's declaration of Helsinki. Written informed consent was obtained from all participants, their parents or statutory guardians.

## Examinations and measurement

Ocular biometry, including AL, corneal curvature radius (CR) and anterior chamber depth (ACD), were measured using an IOLMaster (IOLMaster 500, Zeiss, German) in both eyes at baseline and each follow-up visit. Autorefraction was measured using an autorefractor (KR8800, Topcon Corp, Tokyo, Japan) after cycloplegia (3 drops of 1% cyclopentolate). The biological parents of the twins underwent autorefraction measurements in both eyes without cycloplegia. Height and weight were recorded at baseline and each follow-up visits.

## Definition and statistical methods

Spherical equivalent refraction (SER) was calculated as the sphere plus1/2 cylinder power. Emmetropia was defined as −0.50 D < SER < + 1.0 D. Myopia was defined as an SER ≤ −0.50 Diopters (D). Myopia progression was defined as change in SER greater than −0.25 D/year before age of 15 years. All the subjects were divided into three groups according to the refraction status at baseline and at last visit: persistent emmetropia (PE, −0.50 D < SER < + 1.0 D at baseline and at last visit), incident myopia (IM, −0.50 D < SER < +1.0 D at baseline, SER ≤ −0.5 D at last visit), persistent myopia (PM, SER ≤ −0.5 D at baseline and at last visit). Children with persistent myopia were further divided into progressive myopia (PrM, SER ≤ −0.5 D at baseline and with progression of −0.25 D/year or above), and non-progressive myopia (NPrM, SER ≤ −0.5 D at baseline and with progression less than −0.25 D/year). Age was adjusted as a confounder for axial length by analyzing each age group separately, from 7 to 17 years old, in one-year increments.

Two different methods were utilized to determine the physiological growth of AL. The first method involved assessing the annual AL progression in the NPrM group. The second method was the estimation of compensation AL growth based on the Gullstrand eye model. The calculation used the methods outlined by Mutti et al [17]. Specifically, the following equation for calculating the power of the eye was utilized:

$$\frac{dD}{dmm} = \frac{1336}{H'AL^2}$$

H'AL represents the distance between second principal point and axial length, and the second principal point is 1.6 mm behind the cornea in the Gullstrand No.1 eye [21]. Then the uncompensated AL change can be calculated as:

$$Unompensated\ AL\ change = \left| \frac{SER\ change \times (AL - 1.6)^2}{1336} \right|$$

The compensated AL change can be calculated as:

$$Compensated\ AL\ change = Total\ AL\ change - Unompensated\ AL\ change$$

Right eyes of the first-born twin were arbitrarily selected for analysis. The distribution of age, SER, AL and height were reported with means and standard deviation (SD). Logistic regression models were utilized to classify between PE vs. IM/PM and PrM vs. NPrM. The outcome variable was categorical, where PE was represented as 1 vs. IM/PM as 0, and PrM as 1 vs. NPrM as 0. The predictor variable (x) used in these models was the annual AL

progression. Receiver operating characteristic (ROC) curves were plotted, and the area under the curve (AUC) was calculated using a logistic model. The cutoff point to determine the outcome was selected using the Youden Index method [22]. The Youden Index is a metric that combines sensitivity and specificity into a single measure. It helps in determining the optimal cutoff point by maximizing the vertical distance between the ROC curve and the diagonal line (representing a random classifier). This point balances sensitivity and specificity, offering a threshold that optimizes the trade-off between true positives and false positives. All the analysis was conducted using Stata software (version 18.0, StataCorp, TX, USA).

## Results

### Study population and baseline characteristics

A total of 916 children with a mean age at baseline of $11.62 \pm 2.74$ years were included in the current study. The female gender constituted 53.4% of the population. The baseline characteristics of the groups of PE, IM and PM were summarized in Table 1. Among the three groups, the PM group consisted of the oldest subjects ($12.57 \pm 2.63$ years, $P < 0.001$, oneway ANOVA) and had the highest proportion of females (56.1%, $P < 0.001$, chi-square test). Interestingly, this group also exhibited the longest AL ($23.59 \pm 0.71$ mm, $P < 0.001$, oneway ANOVA).

### Annual axial length changes across refractive error groups

As seen in Table 2, the annual change in AL was found to be significantly smaller in the PE group compared to the IM and PM groups at all ages (all $P < 0.05$, Bonferroni test). However, there was still a growth of 0.14 to 0.20 mm per year in AL from the age of 7 to 11, with minimal changes in SER during this period. There were no statistically significant differences in AL growth between the IM and PM groups at all ages (all $P > 0.05$, Bonferroni test), and both groups exhibited a decrease in growth with increasing age. The difference in SER growth between the IM and PM groups was not significant from the age of 7 to 16 (all $P > 0.05$, Bonferroni test). Changes in height showed no significant differences among the three groups from the age of 7 to 17 (all $P > 0.05$, Bonferroni test).

### Patterns of two types of physiological AL growth

Two types of physiological growth of AL were described in Table 3. Overall, the physiological AL growth calculated using Gullstrand model in the PE, IM and PM groups decreased with age. The PE group shows the smallest change in physiological AL growth, and the PM group has similar values and trend compared with PE group. The PE group exhibits the most

**Table 1. Baseline characteristics of persistent emmetropia, incident myopia, and persistent myopia groups.**

|  | Persistent emmetropia n = 169 | Incident myopia n = 262 | Persistent Myopia n = 485 |
|---|---|---|---|
| **Age, year** | | | |
| Mean+SD | $11.38 \pm 2.76$ | $9.99 \pm 2.05$ | $12.57 \pm 2.63$ |
| Range | 5.03, 17.48 | 6.40,16.33 | 6.95,15.29 |
| **Female, %** | 45.0% | 53.8% | 56.1% |
| **SER at 7 years, D** | $0.67 \pm 0.27$ | $0.27 \pm 0.50$ | $-1.93 \pm 1.52$ |
| **AL at 7 years, mm** | $22.60 \pm 0.61$ | $23.17 \pm 0.61$ | $23.59 \pm 0.71$ |
| **Height at 7 years, cm** | $123.94 \pm 4.34$ | $126.25 \pm 4.25$ | $125.55 \pm 5.89$ |

AL: axial length. SER: spherical equivalent. SD: standard deviation.

**Table 2. Annual change in axial length, refraction and height during 6-year follow-up among different age groups.**

| Age, year | AL mm/y, Mean ± SD | | | SER D/y, Mean ± SD | | | Height cm/y, Mean ± SD | | |
|---|---|---|---|---|---|---|---|---|---|
| | PE | IM | PM | PE | IM | PM | PE | IM | PM |
| 7 | 0.18 ± 0.10 | 0.39 ± 0.16 | 0.43 ± 0.15 | −0.10 ± 0.30 | −0.48 ± 0.39 | −0.73 ± 0.43 | 5.95 ± 1.87 | 5.64 ± 2.16 | 5.77 ± 2.42 |
| 8 | 0.17 ± 0.09 | 0.46 ± 0.20 | 0.44 ± 0.23 | −0.03 ± 0.24 | −0.81 ± 0.50 | −0.82 ± 0.53 | 5.51 ± 1.49 | 5.43 ± 2.68 | 5.19 ± 1.83 |
| 9 | 0.18 ± 0.12 | 0.41 ± 0.19 | 0.44 ± 0.14 | −0.20 ± 0.29 | −0.74 ± 0.48 | −0.89 ± 0.48 | 6.18 ± 1.85 | 6.30 ± 2.12 | 6.49 ± 2.39 |
| 10 | 0.14 ± 0.09 | 0.37 ± 0.18 | 0.38 ± 0.17 | −0.10 ± 0.37 | −0.71 ± 0.41 | −0.76 ± 0.47 | 7.48 ± 1.71 | 6.43 ± 1.91 | 6.26 ± 2.75 |
| 11 | 0.20 + 0.20 | 0.31 ± 0.14 | 0.34 ± 0.14 | −0.24 ± 0.37 | −0.58 ± 0.38 | −0.62 ± 0.43 | 6.77 ± 3.53 | 6.24 ± 2.95 | 6.17 ± 2.74 |
| 12 | 0.08 ± 0.08 | 0.26 ± 0.15 | 0.26 ± 0.14 | −0.20 ± 0.57 | −0.52 ± 0.38 | −0.55 ± 0.43 | 4.99 ± 3.15 | 4.83 ± 2.96 | 4.87 ± 3.36 |
| 13 | 0.12 ± 0.19 | 0.18 ± 0.14 | 0.21 ± 0.12 | 0.00 ± 0.41 | −0.46 ± 0.38 | −0.42 ± 0.38 | 5.11 ± 5.92 | 3.09 ± 2.83 | 3.10 ± 3.24 |
| 14 | 0.10 ± 0.08 | 0.18 ± 0.14 | 0.16 ± 0.12 | −0.13 ± 0.32 | −0.45 ± 0.38 | −0.35 ± 0.33 | 2.08 ± 1.50 | 1.78 ± 1.94 | 1.53 ± 2.06 |
| 15 | 0.03 ± 0.09 | 0.12 ± 0.11 | 0.13 ± 0.14 | 0.04 ± 0.32 | −0.27 ± 0.40 | −0.23 ± 0.40 | 0.85 ± 1.29 | 0.66 ± 1.11 | 0.90 ± 1.43 |
| 16 | 0.04 ± 0.07 | 0.13 ± 0.09 | 0.10 ± 0.10 | −0.04 ± 0.33 | −0.32 ± 0.36 | −0.20 ± 0.35 | 0.55 ± 1.16 | 0.34 ± 1.00 | 0.47 ± 1.44 |
| 17 | 0.04 ± 0.09 | 0.13 ± 0.08 | 0.09 ± 0.11 | −0.07 ± 0.28 | −0.39 ± 0.38 | −0.16 ± 0.34 | 0.22 ± 0.99 | 0.23 ± 1.02 | −0.06 ± 1.07 |

AL: axial length. SER: spherical equivalent refraction. SD: standard deviation. PE: persistent emmetropia. IM: incident myopia. PM: persistent myopia.

**Table 3. The physiological growth of axial length defined by compensation to refractive change (calculated based on the Gullstrand eye model), or myopia progression (annual progression ≤ 0.25 D).**

| Age, year | Compensated axial length mm/year, mean ± SD | | | NPrM mm/year, mean ± SD |
|---|---|---|---|---|
| | PE | IM | PM | |
| 7 | 0.11 ± 0.13 | 0.20 ± 0.11 | 0.15 ± 0.10 | – |
| 8 | 0.10 ± 0.09 | 0.16 ± 0.10 | 0.13 ± 0.12 | 0.19 |
| 9 | 0.08 ± 0.12 | 0.13 ± 0.11 | 0.09 ± 0.11 | 0.23 ± 0.04 |
| 10 | 0.04 ± 0.09 | 0.11 ± 0.13 | 0.07 ± 0.14 | 0.19 ± 0.20 |
| 11 | 0.07 ± 0.20 | 0.09 ± 0.12 | 0.08 ± 0.12 | 0.20 ± 0.10 |
| 12 | −0.03 ± 0.18 | 0.06 ± 0.10 | 0.02 ± 0.11 | 0.17 ± 0.11 |
| 13 | 0.03 ± 0.19 | 0.05 ± 0.10 | 0.02 ± 0.17 | 0.13 ± 0.14 |
| 14 | 0.00 ± 0.08 | 0.00 ± 0.11 | 0.00 ± 0.11 | 0.05 ± 0.11 |
| 15 | −0.06 ± 0.13 | −0.27 ± 0.10 | −0.03 ± 0.14 | 0.12 ± 0.16 |
| 16 | −0.05 ± 0.10 | 0.00 ± 0.12 | −0.02 ± 0.10 | 0.11 ± 0.11 |
| 17 | −0.04 ± 0.12 | −0.05 ± 0.08 | −0.04 ± 0.10 | 0.10 ± 0.10 |

PE: persistent emmetropia. IM: incident myopia. PM: persistent myopia. NPrM: non-progressive myopia SD: standard deviation.

minimal changes in physiological AL growth, while the PM group demonstrates similar values and trends in comparison to the PE group. In contrast, the IM group presents higher physiological AL growth rates than the PE group, reaching a peak at the age of 8, followed by a decline to less than 0.1 mm/year after the age of 12. Regarding the physiological growth of AL calculated based on SER progression, which corresponds to AL progression in the NPrM group, it shows a higher mean AL growth compared to the compensated AL in the PM group across all ages, with this difference being particularly marked at ages 8 and 9. Even after the age of 13, children with less than 0.25 D progression continued to show a growth rate of approximately 0.10 mm/year, whereas the compensated AL growth decreased to around 0 mm/year.

Fig 1 illustrates the comparison of overall AL growth across different refractive error groups with two distinct types of physiological growth. When comparing the AL growth in the PE and IM groups, the growth in the PrM group was similar to the IM group at all ages.

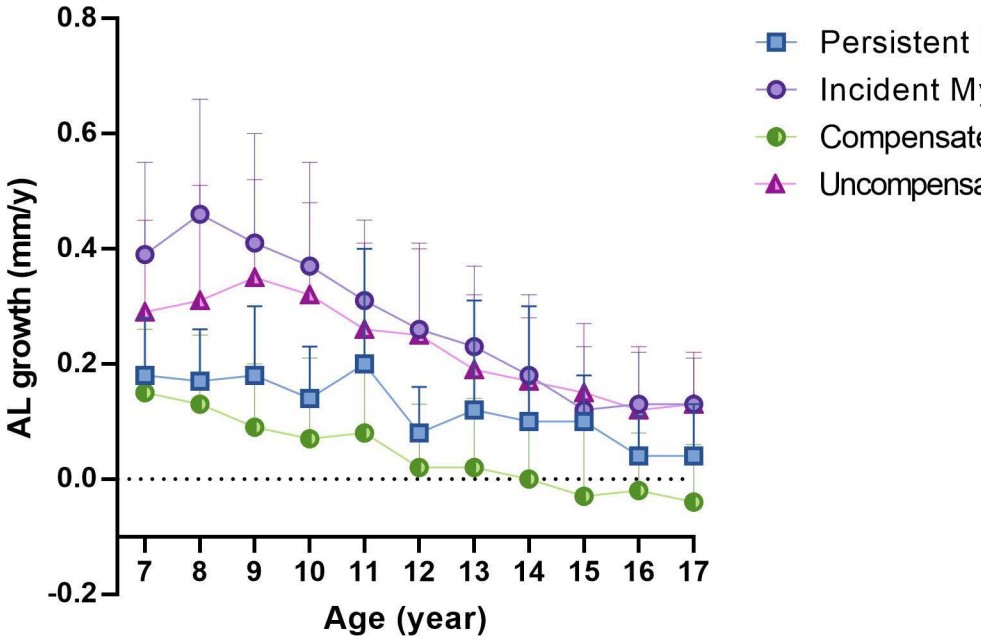

**Fig 1. The annual change of axial length (mm/year) among persistent emmertropia, incident myopia, (A) progressive and non-progressive myopia, and (B) compensated and uncompensated growth.**

In contrast, the physiological growth based on SER progression, which corresponds to the AL growth in the NPrM group, was similar to the PE group at all ages (Fig 1A). However, when using the compensation method to calculate physiological AL growth, the uncompensated AL growth was smaller than the IM group before the age of 9, whereas the compensated AL growth remained consistently smaller than the PE group across all age ranges (Fig 1B).

### Cutoffs for physiological AL growth based on refraction progression

A logistic model was used to further investigate the cutoffs of physiological growth of AL by distinguishing between emmetropia and myopia, as well as between progressive myopia and non-progressive myopia. ROCs were constructed based on the logistic regression models and AUC was calculated. As shown in Table 4, for children aged 7–9 years, the AUC was 0.90 (95%CI 0.85 to 0.94) for the PE vs. IM/PM comparison and 0.87 (95%CI 0.78 to 0.97) for the PrM vs. NPrM comparison. The cutoff point for AL growth to distinguish between PE and IM/PM was determined to be 0.31 mm/year, while it was 0.19 mm/year to differentiate between PrM and NPrM, as determined by the Youden Index method. In the case of children aged 10–14 years, the cutoffs for AL growth were 0.19 mm/year (AUC = 0.78, 95% CI 0.75 to 0.82) for the PE vs. IM/PM comparison and 0.21 mm/year (AUC = 0.80, 95% CI 0.76 to 0.85) for the PrM vs. NPrM comparison. However, for children aged over 15 years, the cutoff figures for AL growth were 0.20 mm/year and 0.04 mm/year, respectively. It is important to note that the AUCs were relatively low in these cases (0.69 [0.64, 0.74] for the PE vs. IM/PM comparison and 0.53 [0.48, 0.58] for the PrM vs. NPrM comparison).

### Discussion

In this study, we compared two different approaches to calculate physiological AL growth in a longitudinal cohort aged 7 to 18 years. We found that the physiological growth calculated

based on refractive error progression closely resembled that of persistently emmetropic eyes. On the other hand, the physiological growth calculated based on whether refractive error was compensated only showed a gradual but low-level increase. The cutoff point for AL growth to distinguish between progression myopia and non-progressive myopia was determined to be 0.19 mm/year. The difference between compensatory AL growth and the growth in non-progressive myopia may have different implications in clinical practice. The value of compensatory AL growth is quite small, and if we base the timing of myopia intervention on this value rather than the traditional −0.75 D threshold, it could lead to unnecessary treatments. On the other hand, the AL growth in non-progressive myopia is closer to the axial length increase corresponding to 0.75 D, making it a useful supplement to the refractive standard. When evaluating the effectiveness of myopia interventions, using the total axial length progression minus the AL growth in non-progressive myopia might underestimate the intervention's efficacy, potentially leading to unnecessary adjustments in treatment plans. Therefore, compensatory AL growth should be used as a reference to estimate the effectiveness of myopia treatments, particularly for orthokeratology lenses.

The observed AL growth in emmetropic eyes in our study aligns with findings from previous studies [23,24]. In a study conducted in South China documented an annual AL growth of 0.16 mm among Grade 1 students, and 0.17 mm among Grade 4 students with persistent emmetropia in elementary school, demonstrating consistency with our results [25]. Rozema et al. reported an increase in AL of 0.12 mm at age 8, 0.17 mm at age 9, and 0.10 mm at age 11 in persistently emmetropic children [26]. Similarly, Fledelius et al. found an annual AL growth of 0.14 mm in emmetropic children at the age of 8–10 years [27]. The pattern of physiological growth in emmetropic eyes from these studies is in consistent with the growth curves plotted by Truckenbrod et al. [28], confirming a slowly increasing AL elongation in emmetropic eyes before adulthood. The difference in axial elongation between emmetropic and myopic eyes in our results, which decreases with age, is inconsistent with the estimation provided by Naduvilath et al., despite their models indicating a higher progression rate in emmetropic eyes [29].

Our results indicate that if physiological growth is calculated using myopia progression < 0.25 D, ranging from 0.13 mm/year to 0.23 mm/year in individuals aged 8–13, the growth rate is similar to that observed in emmetropic eyes. These values are comparable to the AL growth of 0.16–0.18 mm/year among persistent myopia subjects aged 6–8 years, as reported by Chen et al [25]. Compared to the results of the East Asian population from Yii et al., the growth rate from the age of 8 was similar (0.13 mm/year), but their rates decreased to less than 0.10 mm/year after the age of 9 [15]. Our results were lower than the median value of 0.25 mm/year between the ages of 6–10 in the Shanghai cohort [16]. However, their 5th percentile values were closer to the findings of Yii et al., which were 0.11 mm/year at the age of 11

**Table 4. The associations between axial length growth and factors before age of 15 years in generalized linear model.**

|                   | 7–9 year           | 10–14 year         | > =15 y ear        |
|-------------------|--------------------|--------------------|--------------------|
| **PE v.s. IM/PM** |                    |                    |                    |
| **AUC**           | 0.90(0.85, 0.94)   | 0.78(0.75, 0.82)   | 0.69(0.64, 0.74)   |
| **Cutoff, mm/y**  | 0.31               | 0.19               | 0.20               |
| **PrM v.s. NPrM** |                    |                    |                    |
| **AUC**           | 0.87(0.78, 0.97)   | 0.80(0.76, 0.85)   | 0.53(0.48, 0.58)   |
| **Cutoff, mm/y**  | 0.19               | 0.21               | 0.04               |

PE: persistent emmetropia. IM: incident myopia. PM: persistent myopia. PrM: progressive myopia. NPrM: non-progressive myopia. AUC: area under the curve.

and < 0.10 mm/year after the age of 9 [16]. It can be observed that when defining physiological growth based on non-progressive myopia, children aged 6–8 years still exhibit AL growth of 0.1–0.2 mm/year despite minimal refractive changes. For children above the age of 9, there is significant variation among different studies, with some showing minimal or almost no growth, while others report rates exceeding 0.2 mm/year, with the high growth rates accounting for 50% of the population [16]. This indicates that if refraction is the target for controlling myopia, a considerable number of myopic children will continue to experience axial elongation, and the cumulative absolute reduction in axial elongation (CARE) [30] will continue to increase. As seen in our data, although children with non-progressive myopia exhibit minimal refractive changes (−0.09 + 0.15 D) over 6 years, ranging from −0.25 to 0.38 D, the AL growth remains as high as 0.2 mm in children aged 7–11. Therefore, even non-progressive myopia is a commonly used reference in clinical practice, it only reflects the risk of complications in a subset of children aged 9 and above (possibly less than 5%). The cutoff of AL growth for identifying progressive and non-progressive myopia may serve as the minimum requirement for children who have not yet started intervention and must receive it.

We calculated physiological growth using another method proposed by Mutti et al. [17], which essentially predicts changes in eye length caused by refractive progression using a Gullstrand #2 schematic eye. It calculates the eye length that does not contribute to refraction change, known as the compensated length, based on the actual growth of the eye length. Since we lack data on vitreous chamber depth and lens parameters, we used the Gullstrand #1 schematic eye calculate the compensated AL growth. Interestingly, the calculated compensated AL values closely align with Mutti's compensated vitreous chamber depth values in the American population (0.14 mm/year at the age of 7, 0.16 mm/year at 8, and less than 0.1 mm/year after the age of 9). However, in the myopic group (IM group), the compensated change in AL was higher than that observed in myopic eyes in the American population [17]. As for non-myopic eyes, Mutti's study reported a compensated change in vitreous chamber depth of 0.14 to 0.14 mm/year at the age of 7–8, which decreased to 0.059 mm/year at the age of 9. In our study, the compensated AL growth in the progressive emmetropia (PE) group and the progressive myopia (PM) group showed slightly higher but similar patterns compared to the non-myopic eyes in Mutti's study [17]. In other words, after the age of 9, almost all the growth of the AL is reflected in myopia progression, which significantly differs from the AL growth rate calculated based on myopic progression using a 0.25 D cutoff. Therefore, when evaluating the effectiveness of myopia control interventions, we may need to recognize that the compensatory growth of AL in myopic children is minimal. Interventions should aim to maximize the controlling effect of AL growth without considering the reserved compensatory.

The current study demonstrates several strengths that contribute to its reliability and significance. First, it benefits from a long follow-up period, allowing for a comprehensive assessment of AL growth over time. Second, the Guangzhou Twin Eye Study cohort is representative of the general population [31]. Furthermore, the analysis covers a diverse range of refraction statuses, allowing direct comparisons of AL patterns among them. While several limitations should be acknowledged. One limitation is the relatively small number of young children with non-progressive myopia, which could affect the generalizability of findings for this subgroup. Additionally, the absence of data on lens thickness and lens power restricts the precise calculation of compensation in AL growth. A study comparing the predicted AL from the Gullstrand simplified model with the measured AL showed that the predicted AL was generally longer, with the difference being more pronounced in the myopic group compared to the emmetropic group [32]. This suggests that the estimation of compensatory growth in AL may be overestimated in myopic eyes in our study. Our results may need to be validated using

data that includes lens thickness. Therefore, further studies are warranted to fill these gaps in understanding.

## Conclusions

Our findings indicate a notable disparity between the estimates of axial length growth based on myopic progression and compensation calculations. The thresholds for physiological AL growth defined by myopic progression may provide a reference for determining the appropriate timing of interventions, particularly for older children, rather than relying solely on myopic progression itself. Conversely, the AL growth calculated by Gullstrand eye model suggest that the physiological reserve plays a minimal role when estimating the intervention outcome for myopia.

## Supporting information

**S1 Fig. The enrollment and follow-up flow chart of the study population.**
(TIF)

**S2 Data. The raw data of the study.**
(PDF)

## Author contributions

**Conceptualization:** Yanxian Chen, Yangfa Zeng, Mingguang He.

**Data curation:** Xiaohu Ding, Ruilin Xiong, Jian Zhang.

**Formal analysis:** Yanxian Chen, Jian Zhang, Fan Song.

**Investigation:** Ruilin Xiong, Ziwei Zhao.

**Methodology:** Yanxian Chen, Xiaohu Ding, Fan Song.

**Project administration:** Mengying Lai.

**Resources:** Ruilin Xiong, Jian Zhang, Fan Song, Ziwei Zhao.

**Supervision:** Mengying Lai, Yangfa Zeng, Mingguang He.

**Validation:** Ziwei Zhao.

**Writing – original draft:** Yanxian Chen.

**Writing – review & editing:** Yanxian Chen, Xiaohu Ding, Yangfa Zeng, Mingguang He.

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
