## [Decision Letter · Decision Letter 0]

11 Nov 2024

PONE-D-24-41318Physiological growth of axial length among Chinese children and teenagers: A 6-year cohort studyPLOS ONE

Dear Dr. Chen,

Thank you for submitting your manuscript to PLOS ONE. After careful consideration, we feel that it has merit but does not fully meet PLOS ONE’s publication criteria as it currently stands. Therefore, we invite you to submit a revised version of the manuscript that addresses the points raised during the review process.

We look forward to receiving your revised manuscript.

Kind regards,

Clara Martínez Pérez

Academic Editor

PLOS ONE

Journal Requirements:

“PolyU - Rohto Centre of Research Excellence for Eye Care (Collaborative) (P0046333) and Start-up Fund for RAPs under the Strategic Hiring Scheme(P0048638)”

“NO authors have competing interests”

5. We note that your Data Availability Statement is currently as follows: [All relevant data are within the manuscript and its Supporting Information files.]

Please confirm at this time whether or not your submission contains all raw data required to replicate the results of your study. Authors must share the “minimal data set” for their submission. PLOS defines the minimal data set to consist of the data required to replicate all study findings reported in the article, as well as related metadata and methods (https://journals.plos.org/plosone/s/data-availability#loc-minimal-data-set-definition ).

If your submission does not contain these data, please either upload them as Supporting Information files or deposit them to a stable, public repository and provide us with the relevant URLs, DOIs, or accession numbers. For a list of recommended repositories, please see https://journals.plos.org/plosone/s/recommended-repositories .

6. Please include your tables as part of your main manuscript and remove the individual files. Please note that supplementary tables (should remain/ be uploaded) as separate "supporting information" files

Reviewers' comments:

Reviewer's Responses to Questions

**Comments to the Author**

1. Is the manuscript technically sound, and do the data support the conclusions?

Reviewer #1: Yes

Reviewer #2: Partly

2. Has the statistical analysis been performed appropriately and rigorously?

Reviewer #1: Yes

Reviewer #2: Yes

3. Have the authors made all data underlying the findings in their manuscript fully available?

Reviewer #1: Yes

Reviewer #2: Yes

4. Is the manuscript presented in an intelligible fashion and written in standard English?

Reviewer #1: Yes

Reviewer #2: Yes

5. Review Comments to the Author

Reviewer #1: This manuscript investigated an important issue related to the physiological growth of axial length among Chinese children and teenagers, providing insights from a 6-year longitudinal cohort study. While the study offers valuable data and perspectives, several aspects require revision for clarity, depth, and rigor.

1. The differences between the two calculation methods for AL growth need better distinction.

2. Please explain the logistic regression models more clearly, especially decision criteria for cutoff points.

3. Authors should discuss the impact of not having lens thickness data on your findings more thoroughly.

4. Please use consistent terminology and clearer subheadings to improve the flow and clarity of the results.

5. It’s important to tie findings more directly to clinical implications for myopia management strategies.

Reviewer #2: The manuscript titled "Physiological growth of axial length among Chinese children and teenagers: A 6-year cohort study" investigates axial length (AL) growth patterns among Chinese youth, aiming to discern physiological from pathological AL elongation associated with myopia progression. Utilizing annual measurements from a cohort of 916 children, the study highlights variations in AL growth across different refractive statuses (emmetropia, incident myopia, persistent myopia) and explores threshold growth rates that may inform myopia control strategies.

1. It is better to mention the ocular (ocular axial length) in the title to indicate the research aim and objectives well.

1. Including "ocular axial length" in the title would enhance clarity, directly aligning the title with the study’s specific aim and objectives, thereby making the research focus immediately apparent.

2. It is recommended to provide detailed follow-up information, including a flowchart illustrating the participant selection process and any losses to follow-up, as this would strengthen the study's transparency and reproducibility.

3. The manuscript should clarify whether potential confounders or covariates were accounted for in the analysis to control their influence on the outcomes, thus reinforcing the robustness of the findings.

4. A comprehensive description of all statistical methods utilized is essential in the Methods section, allowing readers to fully understand and evaluate the analytical approach.

5. Reporting corrected confidence intervals for primary variables and analyses is advised to enhance the precision and interpretability of the results.

6. To improve the figure illustrating axial length (AL) growth, the use of distinct colors for each group would facilitate reader comprehension and improve visual distinction among data categories.

6. PLOS authors have the option to publish the peer review history of their article (what does this mean? ). If published, this will include your full peer review and any attached files.

**Do you want your identity to be public for this peer review?** For information about this choice, including consent withdrawal, please see our Privacy Policy .

Reviewer #1: No

Reviewer #2: No

---

## [Author Response · Author response to Decision Letter 0]

30 Dec 2024

Response to Editor: Thank you for your careful review and detailed instructions. We appreciate the time and effort you have taken to provide us with such constructive feedback. In response to your comments, we have made the necessary revisions to the format and document information to ensure they align with the journal's guidelines. Additionally, we have uploaded the raw data as requested, which we hope will facilitate a more comprehensive evaluation of our work. Please let us know if there are any further adjustments needed. We are committed to meeting the publication standards and are grateful for your guidance in this process.

Reviewer #1: This manuscript investigated an important issue related to the physiological growth of axial length among Chinese children and teenagers, providing insights from a 6-year longitudinal cohort study. While the study offers valuable data and perspectives, several aspects require revision for clarity, depth, and rigor.

1. The differences between the two calculation methods for AL growth need better distinction.

Response: Thank you for your suggestion. We have added a description in Methods to further explain the two methods of defining physiological AL growth. It now reads:

“Two different methods were utilized to determine the physiological growth of AL. The first method involved assessing the annual AL progression in the NPrM group. The second method was the estimation of compensation AL growth based on the Gullstrand eye model.”

2. Please explain the logistic regression models more clearly, especially decision criteria for cutoff points.

Response: Thank you for your valuable suggestion. The details have been added as following:

“Logistic regression models were utilized to classify between PE vs. IM/PM and PrM vs. NPrM. The outcome variable was categorical, where PE was represented as 1 vs. IM/PM as 0, and PrM as 1 vs. NPrM as 0. The predictor variable (x) used in these models was the annual AL progression. Receiver operating characteristic (ROC) curves were plotted, and the area under the curve (AUC) was calculated using a logistic model. The cutoff point to determine the outcome was selected using the Youden Index method. [22] The Youden Index is a metric that combines sensitivity and specificity into a single measure. It helps in determining the optimal cutoff point by maximizing the vertical distance between the ROC curve and the diagonal line (representing a random classifier). This point balances sensitivity and specificity, offering a threshold that optimizes the trade-off between true positives and false positives.”

3. Authors should discuss the impact of not having lens thickness data on your findings more thoroughly.

Response: We agree with the reviewer and have added the discussion on lens thickness in the limitation part:

“Additionally, the absence of data on lens thickness and lens power restricts the precise calculation of compensation in axial length growth. A study comparing the predicted AL from the Gullstrand simplified model with the measured AL showed that the predicted AL was generally longer, with the difference being more pronounced in the myopic group compared to the emmetropic group.[32] This suggests that the estimation of compensatory growth in axial length may be overestimated in myopic eyes in our study. Our results may need to be validated using data that includes lens thickness.”

4. Please use consistent terminology and clearer subheadings to improve the flow and clarity of the results.

Response: The inconsistency was revised, and subheadings have been added. Thank you!

5. It’s important to tie findings more directly to clinical implications for myopia management strategies.

Response: Thank you for your suggestion. We have added a more detailed description on this in the discussion.

Reviewer #2: The manuscript titled "Physiological growth of axial length among Chinese children and teenagers: A 6-year cohort study" investigates axial length (AL) growth patterns among Chinese youth, aiming to discern physiological from pathological AL elongation associated with myopia progression. Utilizing annual measurements from a cohort of 916 children, the study highlights variations in AL growth across different refractive statuses (emmetropia, incident myopia, persistent myopia) and explores threshold growth rates that may inform myopia control strategies.

1. It is better to mention the ocular (ocular axial length) in the title to indicate the research aim and objectives well.

1. Including "ocular axial length" in the title would enhance clarity, directly aligning the title with the study’s specific aim and objectives, thereby making the research focus immediately apparent.

Response: We have added “ocular” in the title to make it more understandable. Now it reads: Physiological growth of ocular axial length among Chinese children and teenagers: A 6-year cohort study

2. It is recommended to provide detailed follow-up information, including a flowchart illustrating the participant selection process and any losses to follow-up, as this would strengthen the study's transparency and reproducibility.

Response: A flowchart showing detailed follow-up information has been added as supplementary. Thank you for the suggestion!

3. The manuscript should clarify whether potential confounders or covariates were accounted for in the analysis to control their influence on the outcomes, thus reinforcing the robustness of the findings.

Response: The confounder for axial length has been adjusted by analyzing each age group separately. We have added the description in the methods as follows:

“Age was adjusted as a confounder for axial length by analyzing each age group separately, from 7 to 17 years old, in one-year increments.”

4. A comprehensive description of all statistical methods utilized is essential in the Methods section, allowing readers to fully understand and evaluate the analytical approach.

Response: We agree and have added more details in the methods to make it more understandable. Now it reads:

“Right eyes of the first-born twin were arbitrarily selected for analysis. The distribution of age, SER, AL and height were reported with means and standard deviation (SD). Logistic regression models were utilized to classify between PE vs. IM/PM and PrM vs. NPrM. The outcome variable was categorical, where PE was represented as 1 vs. IM/PM as 0, and PrM as 1 vs. NPrM as 0. The predictor variable (x) used in these models was the annual AL progression. Receiver operating characteristic (ROC) curves were plotted, and the area under the curve (AUC) was calculated using a logistic model. The cutoff point to determine the outcome was selected using the Youden Index method. [22] The Youden Index is a metric that combines sensitivity and specificity into a single measure. It helps in determining the optimal cutoff point by maximizing the vertical distance between the ROC curve and the diagonal line (representing a random classifier). This point balances sensitivity and specificity, offering a threshold that optimizes the trade-off between true positives and false positives. All the analysis was conducted using Stata software (version 18.0, StataCorp, TX, USA).”

5. Reporting corrected confidence intervals for primary variables and analyses is advised to enhance the precision and interpretability of the results.

Response: Thank you for your valuable advice. We have added the 95% confidence intervals to all the AUCs. However, due to the small sample size in the age-specific groups, the 95% CIs include negative values. Additionally, the mean ± standard deviation in axial length will be comparable to Mutti’s study, which serves as a major reference for compensated axial length.

6. To improve the figure illustrating axial length (AL) growth, the use of distinct colors for each group would facilitate reader comprehension and improve visual distinction among data categories.

Response: We have used distinct colors to label each group in Figure 1. Thank you!

---

## [Decision Letter · Decision Letter 1]

6 Jan 2025

Physiological growth of axial length among Chinese children and teenagers: A 6-year cohort study

PONE-D-24-41318R1

Dear Dr. Chen,

We’re pleased to inform you that your manuscript has been judged scientifically suitable for publication and will be formally accepted for publication once it meets all outstanding technical requirements.

Kind regards,

Clara Martínez Pérez

Academic Editor

PLOS ONE

Reviewers' comments:

Reviewer's Responses to Questions

**Comments to the Author**

1. If the authors have adequately addressed your comments raised in a previous round of review and you feel that this manuscript is now acceptable for publication, you may indicate that here to bypass the “Comments to the Author” section, enter your conflict of interest statement in the “Confidential to Editor” section, and submit your "Accept" recommendation.

Reviewer #1: All comments have been addressed

Reviewer #2: All comments have been addressed

2. Is the manuscript technically sound, and do the data support the conclusions?

Reviewer #1: Yes

Reviewer #2: Yes

3. Has the statistical analysis been performed appropriately and rigorously?

Reviewer #1: Yes

Reviewer #2: Yes

4. Have the authors made all data underlying the findings in their manuscript fully available?

Reviewer #1: Yes

Reviewer #2: Yes

5. Is the manuscript presented in an intelligible fashion and written in standard English?

Reviewer #1: Yes

Reviewer #2: Yes

6. Review Comments to the Author

Reviewer #1: The authors have done an excellent job revising the manuscript. The improvements and clarifications they’ve made greatly enhance the overall quality and readability of their work.

Reviewer #2: I appreciate the authors' efforts and the revisions they have made. No more comments or revisions are needed

7. PLOS authors have the option to publish the peer review history of their article (what does this mean? ). If published, this will include your full peer review and any attached files.

**Do you want your identity to be public for this peer review?** For information about this choice, including consent withdrawal, please see our Privacy Policy .

Reviewer #1: No

Reviewer #2: No

---

## [Editor Report · Acceptance letter]

PONE-D-24-41318R1

PLOS ONE

Dear Dr. Chen,

I'm pleased to inform you that your manuscript has been deemed suitable for publication in PLOS ONE. Congratulations! Your manuscript is now being handed over to our production team.

Kind regards,

on behalf of

Dr. Clara Martínez Pérez

Academic Editor

PLOS ONE